# In Vitro Evaluation of Antibacterial and Antifungal Activity of Biogenic Silver and Copper Nanoparticles: The First Report of Applying Biogenic Nanoparticles against *Pilidium concavum* and *Pestalotia* sp. Fungi

**DOI:** 10.3390/molecules26175402

**Published:** 2021-09-05

**Authors:** Maryam Bayat, Meisam Zargar, Elena Chudinova, Tamara Astarkhanova, Elena Pakina

**Affiliations:** Department of Agrobiotechnology, Institute of Agriculture, RUDN University, 117198 Moscow, Russia; meisam.za_ir84@yahoo.com (M.Z.); chudiel@mail.ru (E.C.); tamara-ast@mail.ru (T.A.); e-pakina@yandex.ru (E.P.)

**Keywords:** silver nanoparticle, copper nanoparticle, antifungal activity, antibacterial activity, green synthesis

## Abstract

There is increased attention paid to metallic nanoparticles due to their intensive use in various branches of agriculture and biotechnology, such as pest management, nanosensors, gene delivery, seed treatment, etc. There has been growing interest in applying environmentally friendly strategies for synthesizing nanoparticles without using substances which are hazardous to the environment. Biological practices for the synthesis of nanoparticles have been considered as possible ecofriendly alternatives to chemical synthesis. In the present study, we used biogenic silver and copper nanoparticles which were prepared by a previously reported green method. Moreover, the problem of chemical residues, which usually remain along with chemically synthesized nanoparticles and limit their application, was solved by developing such a green synthesis approach. To study the antibacterial activity of silver and copper nanoparticles, *Pseudomonas aeruginosa* was used; for the evaluation of antifungal activity, the pathogenic fungi *Botrytis cinerea*, *Pilidium concavum* and *Pestalotia* sp. were applied. To the best of our knowledge, this study represents the first time that the antifungal impact of a nanoparticle has been tested on *Pilidium concavum* and *Pestalotia* sp. Silver nanoparticles were found to be the more effective antimicrobial agent against all examined pathogens in comparison to copper nanoparticles. Data from such investigations provide valuable preliminary data on silver nanoparticle-based compounds or composites for use in the management of different pathogens.

## 1. Introduction

Metal-based nanoparticles (NPs), especially silver NPs, have been a focus of interest due to their remarkable properties and applications in different fields, such as medicine, optoelectronics, catalysis, chemical sensing, cosmetics, and, primarily, in the health industry as antimicrobial agents [1,2]. Copper NPs are widely used for water treatment and food processing, in the electrical industries because of their conductivity, and in the field of chemistry as a lubricant and catalyst [3].

One of the greatest concerns of recent years has been bactericide- or antibiotic-resistant organisms [4]. A great effort has been made for the development of safe management strategies for bacterial and fungal diseases that pose less danger to human and animals. When developing these strategies, there has been significant focus on overcoming the problems associated with using conventional bactericides or fungicides, which include new bacterial mutations, drug resistance to antibiotics, outbreaks of pathogenic strains, etc. Thus, the development of more efficient antimicrobial compounds and alternative therapeutic approaches is in high demand. Excess use of antibiotics has resulted in a considerable increase in the number of drug-resistant pathogens, which are more pathogenic than wild strain pathogens [4,5]. Scientists are searching for new classes of disinfection systems which are able to act efficiently against these pathogens. It has been previously established that metallic nanoparticles, especially silver and copper nanoparticles, exhibit inhibitory and microbicidal activity at low rates against a broad spectrum of bacteria, viruses, and fungi; they have, therefore, been investigated as antimicrobial agents [6]. As a result of their effectiveness on resistant strains of microbial pathogens, Ag and Cu NPs are currently among the most powerful alternatives to conventional antimicrobial agents and are used in bandages, wound healing ointments or creams, biomedical and surgical devices as disinfectants, textile coatings, and food packaging and storage [1,7]. Biosynthesized metal nanoparticles or nanoparticle-based composites can also be applied for the management of plant diseases against phytopathogens [8]. There is significant demand to develop new antimicrobial agents to overcome the problems of conventional antimicrobial materials, such as microbial resistance, environmental pollution, and so on. 

Another benefit of using NPs as an alternative to conventional chemicals can be illustrated in following example. Strawberries are susceptible to various diseases due to their high nutritional value. To obtain high yields of strawberries, it is necessary to use chemical materials that protect strawberries. Chemical remedies should not be used 1–3 weeks before the berries ripen. Therefore, there is a need to develop alternative protective options against fungal and bacterial diseases. The use of silver-based green nanoparticles is environmentally friendly and relatively safe for human health. This kind of formulation could be effective in protecting strawberries from various microbial diseases [6].

The antimicrobial activity of these nanoparticles is known to be a function of the surface area in contact with the microorganisms. The small size and high surface area-to-volume ratio of these nanoparticles allow them to have a close interaction with microbial membranes and show their bactericidal properties. The size, shape and stability of NPs, as well as their concentration, all affect their antimicrobial activity; in turn, these factors are prominently affected by the reaction conditions of their synthesis [9,10,11]. 

The reason for the antimicrobial activity of silver is the high affinity of its ions (released from Ag NPs) toward sulfur, phosphorus and nitrogen, which bind to thiol and amino groups, disrupt DNA and protein structure and affect cell viability. The other mechanism is the induction of reactive oxygen species (ROS) by Ag NPs; this forms free radicals, leading to oxidative stress and, subsequently, a bactericidal action [4]. The mechanisms of antimicrobial action of silver NPs on microbial cells could be listed as membrane breakage, peptidoglycan damage, oxidative stress, protein and DNA denaturation, ribosome disassembly, enzyme inactivation, disruption of the electron transport chain and proton motive force (4; 5,7). The antimicrobial characteristics of Cu NPs are related to the ions that are released from nanoparticles and their tendency to alternate between oxidation states (Cu [Ι] and Cu [ΙΙ]). To summarize previous studies, the denaturing effect of Cu ions on DNA, proteins and enzymes in microbes gives Cu its antimicrobial properties [12,13]. 

There are two approaches to manufacturing nanoparticles: top-down and bottom-up approaches [14,15]. The top-down approach involves the size reduction of large macroscopic materials through physical and chemical processes to produce particles at the nanoscale level; this includes methods such as mechanical milling (physical), etching (chemical), electro-explosion (thermal/chemical), sputtering (kinetic), and laser ablation (thermal). This approach is not suitable for the large-scale production of nanoparticles, as it is an expensive and slow process. In the bottom-up approach, nanoparticles are produced from atoms, molecules, and smaller particles. Supercritical fluid synthesis, spinning, use of templates, plasma or flame spraying synthesis, sol and sol-gel processes, laser pyrolysis, aerosol-based processes, chemical vapor deposition (CVD), atomic or molecular condensation, and green synthesis are important examples of bottom-up approaches [14,15]. 

In recent years, many different methods have been developed to produce nano-scaled antimicrobial agents, such as Ag and Cu nanoparticles [4,15]. Common chemical and physical methods for the synthesis of Ag and Cu NPs are costly and not environmentally friendly. The use of toxic and harsh chemicals (e.g., sodium borohydride, potassium bitartrate, methoxypolyethylene glycol, and hydrazine) and toxic solvents, as well as the formation of toxic by-products arising from physical and chemical methods of NP synthesis, is hazardous to the environment. Additionally, it may cause the absorption of toxic chemicals on the surfaces of NPs, raising the toxicity problem and limiting the application of synthesized NPs in the medicine and food industries [4,15]. 

Therefore, it is crucial to develop a reliable, cost-effective and eco-friendly procedure for the synthesis of NPs [1,2]. Due to the drawbacks of conventional methods, green methods employing biological systems like plants, microorganisms (such as fungi, bacteria, and algae) and biopolymers are rapidly developing, where inherently benign organic molecules are not hazardous to human health and the atmosphere. The biosynthesis of metal nanoparticles using plants has received more attention as it is facile, economic and ecofriendly and could be a good alternative to chemical and physical methods. Since metal nanoparticles can be synthesized by the reduction of metal ions, plant extracts may act as reducing agents in nanoparticle synthesis [14]. These natural sources contain biological molecules, including polyphenols, terpenoids, flavones, carbohydrates, proteins, alkaloids, alcohol, phenolic acids, etc., and could be the capping or stabilizing agent [16,17]. The mechanism of synthesis could be considered to be the electrostatic interaction of the metal ion and the functional groups of the phytochemicals presented in the plant extract [4,17].

In our earlier research [18], the water extract of an agro-waste material, strawberry leaf, was used for the first time as a natural source of reducing and capping agents to develop a green, cost-effective and safe method for the biosynthesis of silver, copper, and some other metal-based nanoparticles. The unique structure and characteristics of biosynthesized NPs, such as shape, size, crystallinity, chemical composition and surface adsorbents, were studied using different characterization techniques, including UV-Vis spectroscopy, XRD spectroscopy, FESEM analysis, PCC spectroscopy and FT-IR spectroscopy. According to the obtained results, biosynthesized Ag NPs had a spherical shape and crystalline structure, and Cu NPs were amorphous nano-sheets. Metal nanoparticles, such as Ag and Cu NPs, can be used in a wide range of biological applications, including as antibacterial agents, anticancer agents, antioxidants, etc. [14,15]. The potential biotechnological application of these biogenic NPs could be investigated in order to develop new green methods in the production of nanomaterials. Therefore, in the present report the antibacterial and antifungal activities of biogenic Ag and Cu NPs are evaluated. *Pseudomonas aeruginosa,* which can cause disease in plants, animals and human [19], was selected for evaluation of antibacterial activity of these NPs. 

For evaluation of antifungal activity of these NPs, three pathogenic fungi were selectd: *Botrytis cinerea*, as an unspecialized necrotrophic fungal pathogen that attacks over 200 different plant species [20], *Pilidium concavum*, which is an opportunistic pathogen that causes leaf spots and stem necrosis in a wide range of hosts, mainly on strawberry plants [21] and *Pestalotia* sp., which is reported to be infectious for azalea (*Rhododendron* L.) leaves [22]. There are just a few reports carried out on the last two fungi, and there is no report on the antifungal effect of NPs on these fungi. To the best of our knowledge, it is for the first time that the antifungal impact of a nanoparticle is tested on *Pilidium concavum* and *Pestalotia* sp.

## 2. Results and Discussion

### 2.1. In Vitro Evaluation of Antibacterial Activity of Biogenic Ag and Cu Nanoparticles

In this investigation, for the first time, the biosynthesis of NPs was carried out based on silver, copper, iron, zinc and magnesium salts employing an environmentally benign synthetic strategy using strawberry leaf extract. Strawberry leaf extract contains minerals and biomolecules responsible for the biochemical reactions wherein biological molecules react with the metallic precursors, leading to formation of the NPs. 

In vitro experiments were performed to evaluate the antimicrobial susceptibility of microorganisms and the antimicrobial activity of biosynthesized NPs on *P. aeruginosa* bacteria. Dilution methods [23] with slight modifications served as reference methods for these experiments.

According to the results, the inhibitory effect of Ag NPs increased with an increased concentration of nanoparticles. The effective concentration (EC50) was 4 ppm, and the minimum bactericidal concentration (MBC) was 10 ppm. The use of biosynthesized Ag NPs as antimicrobial agents has been reported previously. For example, Punjabi et al. [24] reported that biosynthesized silver nanoparticles with a size of 40 nm were found to show antimicrobial activity against tested strains and drug-resistant clinical isolates of *P. aeruginosa* with a minimum inhibition concentration (MIC) in the range of 1.25–5 mg/mL during a broth dilution method [24]. Comparing our outcomes with the findings of Punjabi et al., our biosynthesized Ag NPs are more effective antibacterial agents against *P. aeruginosa*. However, there is no significant difference between the average size of the Ag NPs reported by Punjabi et al. [24] and our biosynthesized Ag NPs; the great difference in antibacterial activity may arise from the non-spherical shape and polydispersity of their Ag NPs.

Okafor et al. [25] biosynthesized silver nanoparticles using different plant extracts such as aloe, geranium, magnolia and black cohosh extracts and reported that, at concentration of 4 ppm, all had a bacteriostatic effect on *Pseudomonas* compared to the untreated species, based on the OD readings compared to the control [25]. 

Patra and Baek [26] tested biologically synthesized Ag NPs against various Gram-positive and Gram-negative foodborne pathogenic bacteria, including *B. cereus*, *L. monocytogenes*, *S. aureus*, *E. coli*, and *S. typhimurium*, and reported the moderate antibacterial activity of Ag NPs (9.26–11.57 mm inhibition zone). MBC values are 50 ppm for Gram-positive *B. cereus* and *L. monocytogenes*, 25 ppm for Gram-negative *S. aureus*, and 100 ppm for Gram-negative *E. coli* and *S. typhimurium* [26]. 

According to the literature, the possible mechanism of the antibacterial effect of Ag NPs could be illustrated as the interaction of Ag^+^ ions (released from Ag NPs) with the cell membrane of bacteria. This results in significant damage, such as cell membrane damage by accumulation of ions in interstitial spaces, exhaustion of intracellular adenosine triphosphate, disruption of transport systems including ion efflux, interruption of cellular processes such as metabolism and respiration by reacting with molecules, and so on [8,24,27].

Green synthesized copper nanoparticles also demonstrated antibacterial activity against Gram-negative *P. aeruginosa* bacteria. The inhibitory effect of Cu NPs increased with the increase in NP concentration. The effective concentration (EC50) was 2.2 mg/mL and the minimum bactericidal concentration (MBC) was 5 mg/mL, which is in accordance with earlier published results [9,28]. However, the MBC of Cu nanoparticles is greater than the MBC of Ag NPs, indicating the greater antibacterial activity of Ag NPs than Cu NPs.

Ebrahimi et al. [29] tested green synthesized Cu NPs with a particle size of 17-41nm on different bacteria such as *Bacillus cereus, Staphylococcus aureus*, *Escherichia coli* and *Klebsiella pneumoniae* and reported MBC values of 5, 5, 10 and 10 mg/mL, respectively [29]. According to Mandava et al. [12], the minimum inhibitory concentrations (MIC) of green synthesized Cu NPs were determined by the macrodilution method. MIC values were found to be 100, 50, 75 and 75 ppm for the tested bacteria *E. coli*, *S. typhi*, *M. luteus* and *S. mutans*, respectively [12].

Like Ag NPs, the antimicrobial activity of Cu NPs may be due to several factors, such as the permeation of copper ions released from Cu NPs to the bacterial cell membrane. This destroys the morphology of the cell membrane through Cu NPs attaching to the cell wall, resulting into cell death. Copper ions could also be involved in the cross-linkage of DNA molecules of bacteria and the denaturation of proteins [9,28].

### 2.2. In Vitro Evaluation of Antifungal Activity of Synthesized Ag Nanoparticles

*Botrytis cinerea* was selected for testing the antifungal activity of the biosynthesized Ag NPs. The next fungus was *Pilidium concavum* (Desm.) Höhn. To the best of our knowledge, there are not any reports on the antifungal effect of nanoparticles on this fungus. The other fungal pathogen selected was *Pestalotia* sp., on which little research has been carried out; there is no report on the antifungal effect of NPs on this fungus. Results from the present study indicated that Ag NPs have antifungal activity against different pathogenic fungi. 

The antifungal activity of the Ag NPs was tested using two different methods: the agar dilution method and the spore germination method. 

***Agar dilution method.*** The pathogenic fungi *Botrytis cinerea*, *Pilidium concavum* and *Pestalotia* sp. were used in this experiment. Figure 1 compares the control and the treatments with the 1 ppm, 10 ppm and 100 ppm concentrations of Ag NP suspensions. Figure 2 and Figure 3 show the properties of *B. cinerea* and *P. concavum* growth at the age of nine days on oatmeal agar-based culture medium. The images clearly represent the effectiveness of Ag NPs on fungal growth inhibition. Morphological changes were observed in areas of fungal growth in the initial days.

To calculate the growth inhibition percent of *B. cinerea* and *P. concavum* resulting from Ag NP treatment, the control was considered as reference and the equation of “Percent (%) inhibition’’ was used. The calculated results are shown in Table 1 and Table 2 and Figure 1. To obtain percent inhibition, the average radius of the fungus colony was measured on assigned successive days. The findings of this research show that the biosynthesized Ag NPs exhibit antifungal activity against *B. cinerea* and *P. concavum* when compared to the control. The antifungal activity on *Pestalotia* sp. was visible but not measurable (Figure 4). Here, different types of fungi with different structures and morphology showed different responses to Ag NPs. Since the mechanism of the effect of NPs on fungal growth is not entirely clear, it is difficult to determine the reason for the different responses to NPs, as one type is more sensitive than the others.

Considering the results represented in Table 1 and Table 2, the inhibition percentage decreased over time to the end of the experiment; additionally, the inhibition percentage increased in a dose dependent manner. This could be because of the high density of Ag NPs at which the medium was able to saturate and cohere to fungal hyphae to deactivate pathogenic fungi [30]. For *B. cinerea*, the lowest percentage was 5.7% for the 1 ppm concentration on day 9; this is as compared to 28% obtained with 100 ppm on the same day. For *P. concavum*, the lowest percentage was 6.5% for the 1 ppm concentration on the 9th day, and 65.36% attained with 100 ppm on the same day. According to the values of percent inhibition reported in Table 1 and Table 2, it can be concluded that the presence of Ag NPs in the cultures affected the growth of *P. concavum* more than *B. cinerea*. Both fungi showed a higher inhibition effect at 100 ppm concentration of Ag nanoparticles than other concentrations. 

Ag NPs inhibit the growth of various fungi with different levels of intensity. For instance, the growth of *Sclerotinia sclerotiorum* (Lib.) De Bary slows down two times at the concentration of 3.9 ± 0.3 ppm Ag NPs, while the growth of *Alternaria alternata* (Fr.) Keissl slowed down twice only at a concentration of 28 ± 1 ppm [6].

Although the mechanism of the fungicidal effect of Ag NPs is still not clear, it has been suggested that Ag NPs inhibit the budding process due to the formation of pores on the fungal cell membrane, which can lead to cell death. In addition, it has been mentioned that the antimicrobial activity of Ag NPs might be mediated by the formation of free radicals, and free radicals can cause severe damages to the chemical structure of DNA and proteins [31]. Kim et al. [30] reported the inhibitory effect of commercial Ag NPs against *B. cinerea* on PDA. 100% inhibition was achieved in samples treated with Ag NPs at concentration of 100 ppm. 

*Spore Germination Inhibition.* As the number of germinated spores are crucial for fungi to prominently infect tissue, it is very helpful to inhibit germination of spores [32]. Figure 5 demonstrates that Ag NPs could successfully inhibit spore germination of *B. cinerea* in a concentration-dependent manner. Under the laboratory conditions, a volume of 20 µL of spore suspension was placed on agar culture containing different amounts of Ag NPs of 0 (as control) 1, 10, and 100 ppm. The spore germination was completely inhibited at 100 ppm, which is half of the amount which was reported by Min et al. [33].

It was reported that a low concentration of NPs is sufficient for the management of pathogens as it penetrates the cells efficiently. At 100 ppm concentration of Ag NPs, most fungi had a high inhibition effect. This occurred because the density of the solution increased, causing coherence/clumping of fungal hyphae [8,33,34].

### 2.3. In Vitro Evaluation of Antifungal Activity of Biogenic Cu Nanoparticles

The fungi *B. cinerea* was applied to study the antifungal activity of Cu NPs using the agar dilution method. The antifungal effect of Cu NPs on the macroscopic growth of *B. cinerea* at the 9th day is presented in Figure 6. 

Regarding the obtained results, no significant growth inhibitions on *B. cinerea* were observed in the control or in concentrations of 1 ppm and 10 ppm; however, there was a 25 ± 0.3% growth inhibition for 100 ppm-treated samples. The increase in the concentration of biosynthesized Cu NPs to 100 ppm resulted in a decrease in the colony radius and growth inhibition percentage of *B. cinerea*. Few works have reported on the usage of Cu NPs as antifungal materials against *B. cinerea*. According to Mishra et al. [34], since NPs have large surface-to-volume ratio, they can strongly adhere to the cell surface of fungi and directly penetrate into the cell and damage the cell wall, causing inhibition of cell growth and eventually cell death. 

## 3. Materials and Methods

The preparation of plant extract, as well as synthesis and characterization of NPs, are described in our previous works [18,35]. Briefly, the extract was prepared by boiling dried strawberry leaf in distilled water for 1h and then filtering the mixture. Silver NPs were generated by reduction of 0.01 M AgNO_3_ solution under continuous stirring while heating to 70 °C and adding strawberry leaf extract drop by drop. Copper NPs were produced by reduction of 0.01M CuSO_4_.5H_2_O solution under continuous stirring and drop by drop addition of strawberry leaf extract. The synthesized NPs were characterized using different techniques, such as UV-Vis Spectroscopy, XRD, FESEM, EDS, Photon Cross-Correlation Spectroscopy (PCCS) and FT-IR. According to the results of specification, synthesized Ag NPs have spherical shapes with an average size of 50 nm and an average hydrated size of 133 nm. The structure of Cu NPs are like sheets with an average diameter of 180 nm, thickness of 30 nm and an average hydrated size of 795 nm [18].

### 3.1. In Vitro Evaluation of Antibacterial Activity of Ag and Cu Biogenic Nanoparticles

The method of broth dilution was applied for evaluation of the antibacterial activity of the NPs. The bacterium *Pseudomonas aeruginosa*, isolated from potato tubers from the Moscow region, was used for this purpose. Lysogeny broth (LB) liquid medium was prepared for bacterial growth according to the Bertani (Bertani 1951) method, and LB agar medium was prepared by the addition of 3 g agar to 200 mL LB solution before autoclaving. 

The bacteria were cultured for 24 h in the LB medium before being used for tests. Subsequently, 10 μL of this solution and different amounts of NP powder were added to test tubes containing 2 mL of LB medium and incubated for 24 h at room temperature. Bacterial medium with no NPs was considered as the negative control. The volume of 10 μL from the test tube was put in LB agar petri dishes, swabbed by an L-shape rod and incubated for 24 h. The antibacterial activity for each NP was analyzed after taking an image of the plates and counting the number of grown colonies using the “imageJ” software (1.53 e, Wayne Rasband and contributors, National Institutes of Health, Madison, WI, USA).

*EC50*, *MIC and MBC:* EC50, or the half maximal effective concentration, was defined as the statistically derived median concentration of a substance (e.g., pesticide) which produces a certain effect in 50% on the test organisms, such as a bacteria population or fungus mycelium, and causes 50% growth inhibition after a specified exposure time. We calculated the EC50 values using linear curve graphs and mathematical equations [36]. In this report, EC50 is the concentration of NPs at the point of a 50% decrease in CFU. 

The MBC, or minimum bactericidal concentration, was defined as the lowest concentration of an antimicrobial substance which is needed to kill 99.9% of the final inoculum after incubation for 24 h under specific conditions. After broth macrodilution, MBC was determined by sub-culturing samples from tubes yielding a negative microbial growth after incubation on the surface of non-selective agar plates to determine the number of colony forming units (CFU) after 24 h of incubation [37]. 

### 3.2. In Vitro Evaluation of Antifungal Activity of Biogenic Nanoparticles

The pathogenic fungi *Botrytis cinerea* Pers., *Pilidium concavum* (Desm.) Höhn. and *Pestalotia* sp. were used in this study. *B. cinerea* (strain 19MFrR1) was isolated from the roots of strawberries grown in the Moscow region, and identified by cultural and morphological characters and by PCR, followed by sequencing of the species-specific site. *Pilidium concavum* (Desm.) Höhn. (strain 19FrPil1) and *Pestalotia* sp. (strain 19FrPest1) were donated by Yulia Yvetkova (All-Russian Plant Quarantine Center Federal State Budgetary Institution, “VNIIKR” FGBI).

Two methods were used to study the antifungal properties of NPs: 

(I) An agar dilution method with slight modification was applied using oatmeal agar medium. Oatmeal agar was prepared according to Koneman [38], as oatmeal is a source of nitrogen, carbon, protein and nutrients, and agar is the solidifying agent. Then the extract was divided into different volumes and agar was added to each volume (1.5 g per 100 mL solution), autoclaved and allowed to cool. NPs were dispersed in sterile distilled water by sonication for 20 min at 25 °C. Proper amounts from the NPs suspensions with different concentrations were added to the molten agar under the laminar flow hood, swirled, poured into Petri dishes and allowed to become cool and solidify. Also, a NP free solution poured into the Petri dishes and allowed to solidify as control. To achieve reliable results, the experiment was carried out in triplicate.

The strains of *B. cinerea*, *P. concavum* and *Pestalotia* sp. were replicated and grown in culture media before being used in the experiment. Agar disks of mycelium with 1 cm diameter cut from fungal cultures of seven days’ age were put in the center of each Petri dish and incubated at room temperature. The diameter of each colony (the average of the longest and the shortest diameter in mm) was measured every three days. Finally, the inhibition ratio of NPs was calculated by the following equations [39]:(1)$$\mathrm{Percentage}\;\mathrm{growth}\;\mathrm{inhibition}\;=\;\left[\frac{C-T}{C}\;\right]\;\times \;100$$
where ***C*** is the diameter of the control colony and ***T*** is the diameter of the treatment colony.

The fungi were maintained using the periodic replanting technique, which enabled the cultures to survive over short periods of time. This method is based on transferring the growth from the dry or old medium to a fresh one, providing optimum conditions for fungus growth. In this way, the main disadvantages of high risk of contamination and variability of the characteristics of the strains are avoided;

(II) To determine the ability of *B. cinerea* to germinate spores, oatmeal agar medium was used with the addition of different concentrations of silver nanoparticles.

To produce fungal inoculums of *B. cinerea*, isolates were grown on oatmeal agar plates at 25 °C. Conidia were collected by scraping them with a sterile spatula; they were then suspended in sterile distilled water. The spores were counted under a light microscope, and the suspension was diluted with sterile distilled water to obtain the final concentration (4.3 spore per μL). A volume of 20 µL of spore suspension was placed on agar culture containing different amounts of the Ag NPs, including 0 (as control) 1, 10, and 100 ppm. The plates were incubated in laboratory conditions, and the presence of growth was observed. The number of the colonies was reported after observation of visible growth. To obtain reliable results, the experiment was carried out in triplicate.

***Data Analysis.*** The experimental data were statistically investigated using ANOVA and one-tailed unpaired Student’s t-test for significance testing, where *p* < 0.05 was considered significant. Values are presented as the mean ± SD of the three replications in each experiment.

## 4. Conclusions

In this study, we applied Ag and Cu NPs to evaluate their antibacterial and antifungal activity. Ag and Cu NPs have been biosynthesized previously using strawberry leaf extract through a completely green, inexpensive, and benign method without using any toxic chemicals. This study determined the antibacterial activity of biogenic Ag and Cu NPs against pathogenic bacterium *P. aeruginosa*. Additionally, pathogenic fungi *B. cinerea*, *P. concavum* and *Pestalotia* sp. were used in the evaluation of the antifungal activity of Ag NPs. To our knowledge, this study represents the first time that a nanoparticle was used against *P. concavum* and *Pestalotia* sp. fungi for the evaluation of NP antifungal activity. The most growth inhibition was against *P. concavum*, followed by *B. cinerea*. There was also slight inhibition against *Pestalotia* sp. *B. cinerea* was also applied for evaluation of the antifungal activity of Cu NPs, and Cu NPs were shown to inhibit growth of this fungus. According to the results, these nanoparticles have the potential to be used as an antimicrobial agent in antibacterial and antifungal remediation or as an additive in conventional formulations.

## Figures and Tables

**Figure 1 molecules-26-05402-f001:**
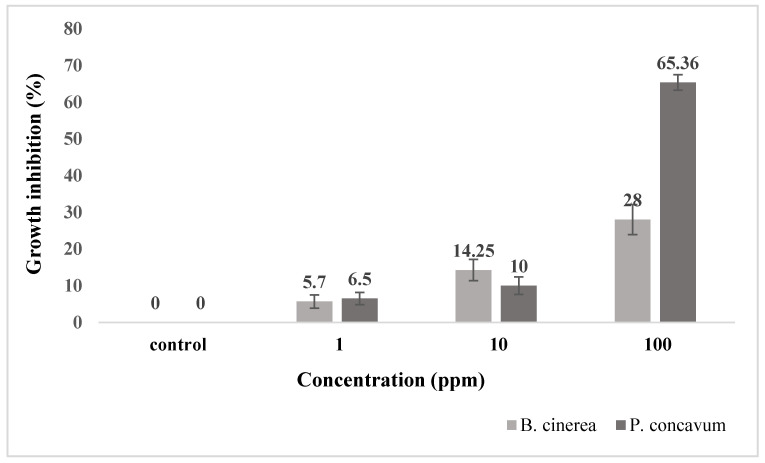
Growth inhibition percentage of *B. cinerea* and *P. concavum* treated with different concentrations of the synthesized Ag NPs for 9 days. Data are presented as mean ± SD of three replications.

**Figure 2 molecules-26-05402-f002:**
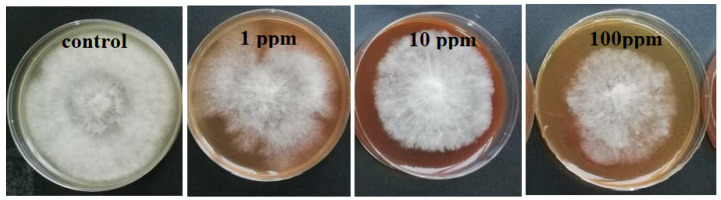
Effect of different concentrations of Ag NP on the growth of *B. cinerea* after 9 days: control and treatments with Ag NPs of 0 (as control) 1, 10, and 100 ppm concentrations, agar dilution method.

**Figure 3 molecules-26-05402-f003:**
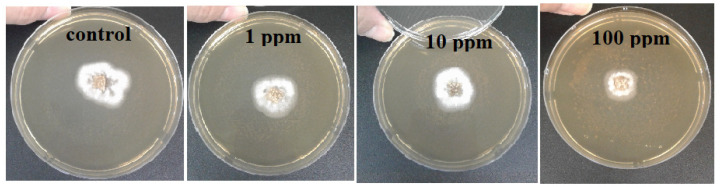
Effect of different concentrations of Ag NP on the growth of *P. concavum* at 9 days of age: control and treatments with Ag NPs of 0 (as control) 1, 10, and 100 ppm concentrations, agar dilution method.

**Figure 4 molecules-26-05402-f004:**
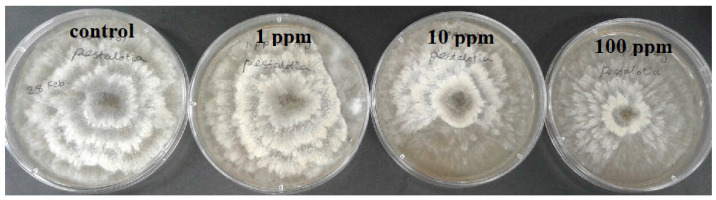
Effect of different concentrations of Ag NP on the growth of *Pestalotia* sp. at 9 days of age: control and treatments with Ag NPs of 0 (as control) 1, 10, and 100 ppm concentrations, agar dilution method.

**Figure 5 molecules-26-05402-f005:**
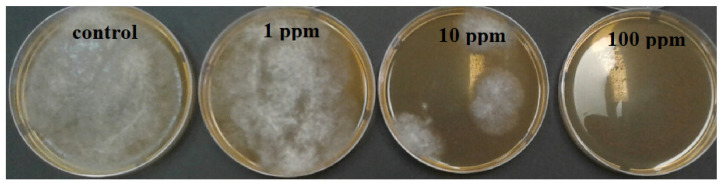
Germination inhibition of *B. cinerea* spores by Ag NPs of 0 (as control) 1, 10, and 100 ppm concentrations dispersed in agar medium.

**Figure 6 molecules-26-05402-f006:**
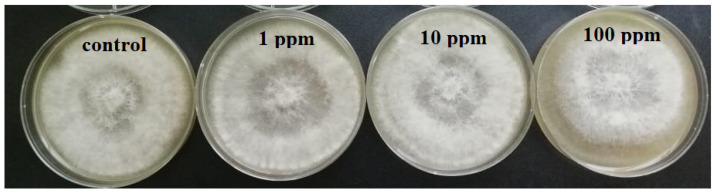
Effect of different Cu NP concentrations on the growth of *B. cinerea* at 9 days of age: control and treatments with Cu NPs of 0 (as control) 1, 10, and 100 ppm concentrations, agar dilution method.

**Table 1 molecules-26-05402-t001:** Growth of *B. cinerea* (colony diameter in mm) treated with different concentrations of biogenic Ag NPs for 9 days. Data are presented as mean ± SD of three replications.

Concentration (ppm)	6th Day Colony Diameter (mm)	9th Day Colony Diameter (mm)	Growth Inhibition % (9th Day)
Control	27.5 ± 0.25	70.5 ± 1.0	-
1	27.3 ± 0.25	66.3 ± 2.2	5.7 ± 1.8%
10	11.9 ± 1.6	60.5 ± 3.2	14.25 ± 2.9%
100	5.0 ± 0.1	50.3 ± 4.9	28 ± 4.1%

**Table 2 molecules-26-05402-t002:** Growth of *P. concavum* (colony diameter in mm) treated with different concentrations of the biogenic Ag NPs for 9 days. Data are presented as mean ± SD of three replications.

Concentration (ppm)	6th Day Colony Diameter (mm)	9th Day Colony Diameter (mm)	Growth Inhibition % (9th day)
Control	3.5 ± 0.25	46.2 ± 6.5	-
1	1.5 ± 0.5	43.2 ± 4.2	6.5 ± 1.7%
10	1.5 ± 0.1	41.6 ± 3.2	10 ± 2.4%
100	0.5 ± 0.1	16 ± 1.0	65.36 ± 2.1%

## Data Availability

The data presented in this study are available in the article.

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
