# Peer review of "In Vitro Evaluation of Antibacterial and Antifungal Activity of Biogenic Silver and Copper Nanoparticles: The First Report of Applying Biogenic Nanoparticles against Pilidium concavum and Pestalotia sp. Fungi"

_molecules, 2021, doi:10.3390/molecules26175402_

Round 1

Reviewer 1 Report

The work can not be published in the journal in the present form. The  editing of English Language is required.  The Figures ( i.e Fig 2) show several problems  such as  3227.33333333.  The Figures should be presente in profesional programs such as Prisma or Origin Graph.  Please, revise the references carefully.  For consistency use the same font size for the Figures.  The work should be considerably improved in order to be resubmitted  to this or other journal. 

Author Response

Dear colleague

We have revised the manuscript for edits/changes as you suggested. All the comments addressed in the text in YELLOW.

We are eagerly waiting to hear from you if any other comments you may have on our manuscript.

-Author name Elena Chudinova was missed and now it’s added.

-The title changed by adding some words to bold the novelty of the work.

Reviewer 1-Comments and Suggestions for Authors

English quality improved

-The Figures ( i.e Fig 2) show several problems  such as  3227.33333333.  The Figures should be presente in profesional programs such as Prisma or Origin Graph. 

Revised

-Please, revise the references carefully. 

Revised

-For consistency use the same font size for the Figures. 

They are in the same font

The work should be considerably improved in order to be resubmitted  to this or other journal. 

 Reviewer2-Comments and Suggestions for Authors

This manuscript describes “In-vitro evaluation of antibacterial and antifungal activity of biogenic silver and copper nanoparticles”.  This is  an interesting work basically explored biogenic silver and copper nanoparticles. Synthesis metallic NP using environmentally benign procedure is interesting research area and needs to explore. Synthesized biogenic Silver and copper NP are exhibiting antibacterial and antifungal activity. This is interesting work and can be consider for publication. Still, there are many shortcomings that will preclude its publication in the current form.

Major and minor concerns:

  1. Introduction of this manuscript very vague. They mostly talked about NPs but they didn’t discuss anything about why their antibacterial and antifungal application is important. Why there is a need to test NPs against tested bacterial and fungal pathogens.

The importance of antibacterial and antifungal application of could be found in first, second and third paragraph of the introduction. Information about tested fungi added to the last paragraph of introduction.

  1. Also, authors need to clarify problem statement about research and briefly discussed research area, and also needs to build a background and rationale about this research with proper references.

It could be found in different parts of the article such as abstract, introduction and results and discussion part, however, if it is so necessary we could repeat it again.

  1. In page 2, At various places this manuscript is not properly cited with relevant references. Sometime author claiming big things without any proper references as in this sentence “In recent years, many different methods have been developed to produce nanoscaled antimicrobial agents like Ag and Cu nanoparticles.” I would recommend authors to cite proper references in introduction.

Revised

  1. As authors claimed that these NPs are first time tested against this fugus “Pilidium concavum” but they didn’t discuss anything about this strain in introduction. Author should discuss about the fungal strains against which these NPs are active. Why issues related to these strains needs to be

The information added to the last paragraph of introduction.

  1. Although, preparation of plant extract, synthesis and characterization of NPs are on previous works. If these NPs are synthesized in different batches, authors need to provide full characterization data of NPs in SI file or in manuscript.

The applied nanoparticles are exactly those which were synthesized in our previous work and the batches are the same, so we referred to our previous work.

Reviewer 3-Comments and Suggestions for Authors

  1. The authors should clearly state in the introduction section why the bacteria Pseudomonas aeruginosa and the fungi Botrytis cinerea, Pilidium concavum and Pestalotia sp are important for the evaluation of their bactericidal and fungicidal effects respectively.

The information added to the last paragraph of the introduction.

  1. Although the authors argue that biogenic nanoparticles are "environmentally friendly" because they do not generate toxic by-products during the synthesis process, some studies have pointed out that biogenically synthesized silver nanoparticles are not necessarily less toxic than those obtained by other processes, such as catalysis. Can you elaborate down this aspect? On the one hand there is the synthesis method, but on the other hand the toxicity and biosafety of the synthesized nanoparticles.

In this paper, "environmentally friendly" mainly refers to the method and in the case of NPs, it is referred due to the residues which are attached to the NPs. The FT-IR spectrum of NPs reveal the existence of organic residues on the surface of NPs [reference 18].

3 Although the physicochemical properties of the experimental nanomaterials used are duly referenced, it is necessary to present in this paper the characterization of both nanoparticles (shape, size, hydrodynamic radius, coating, etc.), to facilitate their contrast with other studies.

Added to the first paragraph of materials and methods part.

  1. It is not necessary to separate the results of Figure 1 and 2 separately, if they are joined together, a better reading of the findings can be achieved.

The two parts of In vitro evaluation of antibacterial activity of biogenic Ag and Cu nanoparticles merged together.

  1. They comment in the paragraph "...and hydrazine), using toxic solvents (e.g. sodium dodecyl benzyl sulphate and polyvinyl pyrrolidone), and formation of toxic by-products arise from physical and chemical methods of NP synthesis will be hazardous to the environment.  ", however they omit to comment that the PVP monomer is toxic, but not the polymer which is widely used in biomedical applications. They need to correct.

Corrected

  1. Another question is about the advantages of "green synthesis", it is not yet known whether biogenically obtained nanoparticles are actually harmless by the synthesis method alone, as this is a notion that has not been clearly demonstrated.

The green synthesis method is a method which use ecofriendly chemicals and procedures. For example, for synthesis of Ag NPs, we can choose a green method or no. in green synthesized NPs, there is not harmless residues. We didn’t discuss about the safety of NP, but about the method, byproducts and residues which are conjugated with NPs.

Reviewer 2 Report

This manuscript describes “In-vitro evaluation of antibacterial and antifungal activity of biogenic silver and copper nanoparticles”.  This is  an interesting work basically explored biogenic silver and copper nanoparticles. Synthesis metallic NP using environmentally benign procedure is interesting research area and needs to explore. Synthesized biogenic Silver and copper NP are exhibiting antibacterial and antifungal activity. This is interesting work and can be consider for publication. Still, there are many shortcomings that will preclude its publication in the current form.

Major and minor concerns:

  1. Introduction of this manuscript very vague. They mostly talked about NPs but they didn’t discuss anything about why there antibacterial and antifungal application is important. Why there is a need to test NPs against tested bacterial and fungal pathogens.
  2. Also, authors need to clarify problem statement about research and briefly discussed research area, and also needs to build a background and rationale about this research with proper references.
  3. In page 2, At various places this manuscript is not properly cited with relevant references. Sometime author claiming big things without any proper references as in this sentence “In recent years, many different methods have been developed to produce nanoscaled antimicrobial agents like Ag and Cu nanoparticles.” I would recommend authors to cite proper references in introduction.
  4. As authors claimed that these NPs are first time tested against this fugus “Pilidium concavum” but they didn’t discuss anything about this strain in introduction. Author should discuss about the fungal strains against which these NPs are active. Why issues related to these strains needs to be
  5. Although, preparation of plant extract, synthesis and characterization of NPs are on previous works. If these NPs are synthesized in different batches, authors need to provide full characterization data of NPs in SI file or in manuscript.

Author Response

(The authors gave the same response as above.)

Reviewer 3 Report

1. The authors should clearly state in the introduction section why the bacteria Pseudomonas aeruginosa and the fungi Botrytis cinerea, Pilidium concavum and Pestalotia sp are important for the evaluation of their bactericidal and fungicidal effects respectively.

2. Although the authors argue that biogenic nanoparticles are "environmentally friendly" because they do not generate toxic by-products during the synthesis process, some studies have pointed out that biogenically synthesized silver nanoparticles are not necessarily less toxic than those obtained by other processes, such as catalysis. Can you elaborate down this aspect? On the one hand there is the synthesis method, but on the other hand the toxicity and biosafety of the synthesized nanoparticles.

3 Although the physicochemical properties of the experimental nanomaterials used are duly referenced, it is necessary to present in this paper the characterization of both nanoparticles (shape, size, hydrodynamic radius, coating, etc.), to facilitate their contrast with other studies.

4. It is not necessary to separate the results of Figure 1 and 2 separately, if they are joined together, a better reading of the findings can be achieved.

5. They comment in the paragraph "...and hydrazine), using toxic solvents (e.g. sodium dodecyl benzyl sulphate and polyvinyl pyrrolidone), and formation of toxic by-products arise from physical and chemical methods of NP synthesis will be hazardous to the environment.  ", however they omit to comment that the PVP monomer is toxic, but not the polymer which is widely used in biomedical applications. They need to correct.

6. Another question is about the advantages of "green synthesis", it is not yet known whether biogenically obtained nanoparticles are actually harmless by the synthesis method alone, as this is a notion that has not been clearly demonstrated.

Author Response

(The authors gave the same response as above.)

Round 2

Reviewer 1 Report

The work was improved. However cannot be yet approved in the present form. Note that the figures need to be improved i.e Figures 2, 3 and 4.  Revise letters size and consistency.

Author Response

Answers

Concentration units of mg/ml changed to ppm to have round figures. For example 0.001mg/ml changed to 1ppm

Reviewer 1

The letters size within the pictures unified.

Reviewer 2

The authors prefer the new title. If you insist, please let us know.

The contents related to the problem statement is highlighted

 in introduction part.

Reviewer 2 Report

Reviewer comments

Major and minor concerns:

  1. Changed Title after revision is weird. Revised title should not be used.
  2. Novelty of this manuscript is low. As Authors just utilized previously synthesized NPs from other manuscript and characterization of these particles is not up to the mark. Authors just tested already synthesized NPs in different strains.
  3. Problem Statement, as authors clarified in comments that “It could be found in different parts of the article such as abstract, introduction and results and discussion part, however, if it is so necessary we could repeat it again.” This is not the way to present research problem. Authors need to describe this with proper references.

Author Response

(The authors gave the same response as above.)

Reviewer 3 Report

Dear authors:
I appreciate your corrections to my remarks. It will always be important to use precise language and to avoid confusion, it should be specified that "green synthesis" is not the same as "environmentally safe nanoparticles". I insist.  For a scientific article must be very clear in its statements.

Author Response

(The authors gave the same response as above.)
